# Novel Fluorescent Probe Based on Rare-Earth Doped Upconversion Nanomaterials and Its Applications in Early Cancer Detection

**DOI:** 10.3390/nano12111787

**Published:** 2022-05-24

**Authors:** Zhou Ding, Yue He, Hongtao Rao, Le Zhang, William Nguyen, Jingjing Wang, Ying Wu, Caiqin Han, Christina Xing, Changchun Yan, Wei Chen, Ying Liu

**Affiliations:** 1Jiangsu Key Laboratory of Advanced Laser Materials and Devices, School of Physics and Electronic Engineering, Jiangsu Normal University, Xuzhou 221116, China; 2020191044@jsnu.edu.cn (Z.D.); 2020201155@jsnu.edu.cn (Y.H.); 2020211255@jsnu.edu.cn (H.R.); zhangle@jsnu.edu.cn (L.Z.); jin16217@mail.ustc.edu.cn (J.W.); wuying@jsnu.edu.cn (Y.W.); hancq@jsnu.edu.cn (C.H.); yancc@jsnu.edu.cn (C.Y.); 2Department of Physics, The University of Texas at Arlington, Arlington, TX 76019-0059, USA; william.nguyen@uta.edu (W.N.); christina.xing@uta.edu (C.X.); 3Medical Technology Research Centre, Chelmsford Campus, Anglia Ruskin University, Chelmsford CM1 1SQ, UK

**Keywords:** upconversion nanomaterials, novel fluorescent probe, miRNA-155, miRNA-150

## Abstract

In this paper, a novel rare-earth-doped upconverted nanomaterial NaYF_4_:Yb,Tm fluorescent probe is reported, which can detect cancer-related specific miRNAs in low abundance. The detection is based on an upconversion of nanomaterials NaYF_4_:Yb,Tm, with emissions at 345, 362, 450, 477, 646, and 802 nm, upon excitation at 980 nm. The optimal Yb^3+^:Tm^3+^ doping ratio is 40:1, in which the NaYF_4_:Yb,Tm nanomaterials have the strongest fluorescence. The NaYF_4_:Yb, Tm nanoparticles were coated with carboxylation or carboxylated protein, in order to improve their water solubility and biocompatibility. The two commonly expressed proteins, miRNA-155 and miRNA-150, were detected by the designed fluorescent probe. The results showed that the probes can distinguish miRNA-155 well from partial and complete base mismatch miRNA-155, and can effectively distinguish miRNA-155 and miRNA-150. The preliminary results indicate that these upconverted nanomaterials have good potential for protein detection in disease diagnosis, including early cancer detection.

## 1. Introduction

MicroRNAs (miRNAs) are important regulators of cell proliferation, division, differentiation, and apoptosis. They can regulate the expression levels of various genes in DNA post-transcriptionally. The abnormal expression of miRNAs has been associated with many diseases (cancer, tumors, and diabetes) [1], and miRNAs can be obtained through blood, urine, etc. [2], which has the advantage of being non-invasive. Therefore, miRNAs are considered important biomarkers for tumors [3]. Patryk Krzeminski [4] et al. demonstrated that DNA methylation contributes to miRNA-155 expression, and the survival data of myeloma cells show a correlation between miR-155 expression and multiple myeloma outcomes. A number of studies have shown that miRNA-155 is closely related to MM, and the overexpression of miRNA 155 in blood is an important signal for the diagnosis of MM [5,6,7]. Therefore, the detection of miRNA-155 to diagnose early MM is an important and efficient method.

The concept of upconversion luminescence was first proposed by Auzel and Ovsyankin et al. [8]. The radiation process is a nonlinear anti-Stokes emission, excited by the effective absorption of two or more low-energy photons. Then, it transitions from the ground state to the excited state through a multi-step process, and finally, it returns to the ground state energy level in radiative transition, realizing the conversion of long-wavelength excitation light into short-wavelength emission light. The excitation wavelength (980 nm) of upconverted nanomaterials is in the “optical transmission window” of biological tissues [9,10,11,12] due to the least light absorbed by biological tissues [13,14,15,16]. This special luminescence process makes upconversion nanomaterials have incomparable advantages over traditional organic dyes, quantum dots, fluorescent proteins, and other biomolecular markers [17,18,19], such as near-infrared light as excitation light, resulting in less photo-damage, lower auto-fluorescence background of biological tissue, and a deep penetration depth [9,10,11,12,13,14,15,16,17,18,19]. In addition, rare-earth-doped upconversion nanoparticles have multi-wavelength emissions, high-fluorescence intensity and high stability, good water solubility, as well as good biocompatibility [20,21,22,23,24,25,26,27,28,29,30,31]. Moreover, rare-earth-doped nanoparticles also have strong scintillation luminescence that can be used for X-ray-induced photodynamic therapy, which is a well-captivated area, as this new therapy can be used for deep as well as skin cancer treatment [20,21,22], radiation dosimetry [23,24] and temperature sensing [25]. These irreplaceable advantages establish upconversion luminescent nanomaterials as a potential fluorescent marker in biological detection [28,29,30,31] and have developed rapidly. For example, Mao et al. [32] prepared NaYF_4_:Yb,Er upconversion nanoparticles using the hydrothermal method, and discovered their specific detection of miRNA by using the principle of base stacking on the surface. Kowalik et al. [33] linked IgG antibodies to the PEG-NHS-modified NaYF_4_:Yb,Tm@SiO_2_ surface to achieve specific labeling to demonstrate the great potential of photodynamic targeted therapy.

The main body of rare-earth-doped upconversion nanoparticle materials is composed of host materials, sensitizers, and activators [34]. The commonly used host materials themselves have no fluorescence but can provide a suitable crystal field for the activated ions, so that the luminescent centers produce specific emissions. Y^3+^, Gd^3+^, Lu^3+^, La^3+^ are usually selected as host elements [35]. The high-quality host material plays a decisive role in the entire upconversion luminescence efficiency. The spacing between rare-earth ions, the symmetry distribution, and the number of coordination ions would affect the crystal structure of the host material. The most common NaYF_4_ is currently recognized as the most effective blue-green light upconversion luminescent host material [36,37]. Because the hexagonal NaYF_4_ has a higher symmetry than the cubic NaYF_4_, the luminous efficiency of the upconversion in hexagonal NaYF_4_ is 10-times higher than in the cubic NaYF_4_ [38]. Nd^3+^ and Yb^3+^ ions are the most common sensitizers in the upconversion systems [39,40]. Yb^3+^ has a large absorption cross-section at 950–1000 nm, which is related to the only excited state and can absorb the excited infrared photons as well as effectively obtain the energy. The energy is then transferred to the activators. Yb^3+^ is more often chosen as the sensitizer of nanomaterials because its upconversion luminescence efficiency is higher than that of Nd^3+^. The activators are rare-earth ions, such as Er^3+^, Tm^3+^, Ho^3+^, Nd^3+^, Pr^3+^, etc. [41,42,43,44], among which the Yb^3+^/Er^3+^ (Tm^3+^, Ho^3+^) pairs are recognized as the most efficient [45].

The standard preparation methods of rare-earth-doped upconversion nanoparticles include the co-precipitation method, high-temperature thermal decomposition method, solvothermal method, microemulsion method, sol–gel method, etc. Nanoparticles with tunable morphology, high-fluorescence intensity, and good dispersion have advantages and disadvantages. For example, Gao et al. [46] compared the effects of factors, such as the concentration of reactants, the ratio of reactants and ligand solvents, and the types of ligand solvents on the synthesis of nanoparticles, respectively. Hydrothermal synthesis has been widely used for NaYF_4_:Yb,Er nanoparticle synthesis. This method has a fast reaction and precipitation rate and crystallinity; however, the fluorescence efficiency of the nanoparticles synthesized is relatively low. Amphiphilic ligands can help the particles to be well dispersed in various polar solvents. However, the surface of the obtained upconversion nanoparticles is usually coated with hydrophobic organic ligands, such as oleic acid, oleylamine, and octadecene, which make them less water soluble and difficult for biological applications. To improve the water solubility and biocompatibility, the surface of upconvertion nanoparticles must be modified with bioactive ligands or biomolecules. Surface ligand exchange, surface ligand oxidation, and surface ligand assembly are common surface modification methods [47,48,49]. Jiang et al. [48] used polymeric anhydride to interact with octadecene and oleic acid ligands on the surface of nanoparticles by coordination adsorption, and then cross linked with dicyclohexanetriamine to obtain stable water-soluble upconverted nanoparticles. The effective modification of upconverted materials has become a key factor in their applications in biological detections.

In this paper, high-temperature thermal decomposition was used to prepare NaYF_4_:Yb^3+^/Tm^3+^ upconverted nanoparticle materials and it was found that the optimal Yb^3+^/Tm^3+^ ratio is 40:1, in terms of the luminescence efficiency. These new types of upconverted nanoparticles were modified by surface coating and tested for protein detection for the purpose of early cancer diagnosis.

## 2. Experimental Materials and Methods

### 2.1. Reagents and Instruments

Chemicals and Reagents: These chemicals were purchased from Shanghai Aladdin Biochemical Technology Co. Ltd. (Shanghai, China): yttrium chloride hexahydrate (YCl_3_ 6H_2_O), ytterbium chloride hexahydrate (YbCl_3_ 6H_2_O), thulium chloride hexahydrate (TmCl_3_●6H_2_O) gadolinium chloride hexahydrate (GdCl_3_ 6H_2_O), oleic acid (OA), 1-octadecene (ODE), ammonium fluoride (NH_4_F), cyclohexane (C_6_H_12_), methanol (MeOH). These Chemicals were purchased from Sinopharm Chemical Reagent Co., Ltd. (Shanghai, China): hydrochloric acid (HCL), succinic acid (C_6_H_10_O_4_), sodium carbonate (Na_2_CO_3_), sodium bicarbonate (NaHCO_3_), acetonitrile (C_2_H_3_N), and 1-ethyl-(3-Dimethylaminopropyl) carbonate diimide hydrochloride (EDC).

Anhydrous ethanol (C_2_H_6_O) and sodium hydroxide (NaOH) were purchased from Xilong Chemical Co., Ltd. (Guangzhou, Guangdong, China). Bovine serum albumin (BSA), N,N-dimethylformamide (DMF), and N-hydroxysuccinimide (NHS) were purchased from Shanghai Macklin Biochemical Co. Ltd. (Shanghai, China). Nitronium tetrafluoroborate (NOBF_4_) was purchased from Hebei Bailingwei Superfine Materials Co., Ltd. (Langfang, Hebei, China), PBS buffer and DNA with amino and FAM were purchased from Sangon Biotech Co. Ltd. (Shanghai, China), corresponding base sequence is 5′-NH2-CCCCCCCCCCCCACCCCTATCACGATTAGCATTAA-6-FAM-3′. The ultrapure water was produced by a water purification system (H20BASIC-B, Sartorius, Germany).

Instrumentation: the intelligent digital magnetic stirring electric heating mantle (ZNCL-TS-250 mL, Shanghai Anchun Instrument Co., Ltd., Shanghai, China) was used to prepare upconverted nanomaterials and carboxylated proteins. Fluorescence was measured using a fluorescence spectrometer (F-4600, Hitachi, Hitachi, Ltd., Tokyo, Japan). Phase analysis of upconverted nanomaterials was carried out by X-ray powder diffractometer (X-ray Diffraction, XRD, D8 ADVANCE, Bruker, Germany). The morphology of the upconverted nanomaterials was characterized by field emission scanning electron microscope (Scanning Electron Microscope, SEM, SU8010, Hitachi Co., Ltd., Tokyo, Japan) and scanning transmission electron microscope (Transmission Electron Microscopy, TEM, FEI TECNAI G2 F20, Hitachi Co., Ltd., Tokyo, Japan). A Fourier transform infrared spectrometer measured infrared absorption (Tensor 27, Bruker, Germany). The luminescence was measured with a 980 nm fiber laser (BOT980-5W, Xi’an Leize Electronic Technology Co., Ltd., Xi’an, China).

### 2.2. Preparation of Upconverted Nanomaterials

Rare-earth-doped upconverted luminescent nanomaterials NaYF_4_:Yb,Tm were synthesized by high-temperature thermal decomposition. The preparation of 2 mmol NaYF_4_:20% Yb^3+^, 0.5% Tm^3+^ nanomaterials was performed by first charging 1.39 mmol YCl_3_ 6H_2_O, 0.6 mmol YbCl_3_ 6H_2_O, and 0.01 mmol TmCl_3_ 6H_2_O into a three-neck round-bottom flask. Then, oleic acid (12 mL) and octadecene (30 mL) were added, and the flask flowed with nitrogen gas for 10 min to ensure no oxygen in the flask. Next, under nitrogen protection with magnetic stirring, the mixture was heated to 160 °C and reacted for 1 h to obtain a pale-yellow solution. The mixture was cooled to 50 °C and then 10 mL of a methanol solution containing 8.0 mmol of ammonium fluoride and 5.0 mmol of sodium hydroxide was added dropwise to the mixture. The reaction was continuously stirred at 50 °C for 30 min to ensure a complete integration. Then, the temperature was raised to 80 °C to evaporate the methanol. During the evaporation of methanol, the solution continued to bubble, and the mixture was continuously heated to 120 °C for 30 min until no more bubbles were generated in the solution. Finally, the sample was heated to 300 °C for 90 min. After the reaction was over, the solution was naturally cooled to room temperature and cyclohexane was added to disperse the mixture. The mixture was centrifuged at 8000 rpm/min for 5 min to obtain a precipitate, which was then washed with cyclohexane. The above steps were centrifuged and washed three times, and the NaYF_4_:Yb,Tm upconverted nanomaterial finally obtained was dispersed in cyclohexane and stored. In addition, the experimental steps for the preparation of NaGdF_4_:Yb,Tm upconverted nanomaterials are the same as above. The YCl_3_·6H_2_O in the experimental material was replaced by GdCl_3_·6H_2_O, and the experimental steps were repeated to obtain NaGdF_4_:Yb,Tm upconverted nanomaterials.

### 2.3. Water-Soluble Upconverted Nanomaterials

Two mL of the cyclohexane solution of the upconverted nanomaterial prepared in the above experiment was prepared and ultrasonically dispersed for 5 min. Twenty mg of NaBF_4_ was dissolved in two mL of acetonitrile solution and was added to the fully dispersed cyclohexane solution of upconverted nanomaterials by stirring at 1000 rpm/min for 30 min to obtain a mixed solution of water and oil separation. Then, the water-soluble NaYF_4_:Yb, Tm upconverted nanomaterials were obtained by centrifuging at 8000 rpm/min for 15 min.

### 2.4. Preparation of Carboxylated Proteins

First, we dissolved 1 g of bovine serum albumin in 20 mL of ultrapure water, then excess oxalic acid was added, and we adjusted the pH to 7–8 with an aqueous sodium carbonate solution under magnetic stirring. Then, 5 mmol of EDC was added and the mixed solution was stirred overnight. A dialysis bag with a molecular-weight cut-off of 10k–30k Da was used for dialysis. The denatured proteins were put into the dialysis bag, clamped on both sides with dialysis clips to prevent leaking, and immersed into a sodium bicarbonate aqueous solution (1000 mL, 2 mmol) leaving it in a refrigerator at 4 °C. The sodium bicarbonate aqueous solution was replaced every 4–6 h to ensure that the protein was always in a slightly alkaline environment. The protein was dialyzed and purified under this condition for at least 72 h, and the carboxylated bovine serum albumin was obtained, which was divided into centrifuge tubes and stored in a −20 °C refrigerator.

### 2.5. Carboxylated Protein-Modified Upconverted Nanoparticles

Next, 0.5 mL of the prepared carboxylated protein, described above, was dissolved in 2 mL of DMF solution, then water-soluble upconverted nanomaterials were added to the DMF solution and the pH was adjusted to about 8 with aqueous sodium bicarbonate solution by stirring for 2 h. After the reaction was completed, the mixed solution was centrifuged at 10,000 rpm/min for 10 min. The precipitates were washed twice with DMF centrifugation, and finally dispersed with DMF for preservation. The above operations were all carried out at room temperature.

### 2.6. DNA Probes Linked to Carboxylated Protein-Modified Upconverting Nanoparticles

The pH of 1 mL of the carboxylated protein-modified upconverted nanomaterials obtained in the above steps was adjusted to about 6 with 20 μmol of hydrochloric acid aqueous solution, then 5 mg EDC and 5 mg NHS were added and stirred at room temperature for 25 min to let the carboxyl groups fully react with the nanoparticles. Next, the aminated DNA probe with FAM was added to a 40 μL PBS buffer to make a concentration of 100 μmol. An appropriate amount of the solution was immediately added to the carboxylated upconverted nanomaterial solution, and then put in a shaker to react for 30 min. The reaction was washed twice with PBS buffer at 10,000 rpm/min for 5 min each time, and the resulting precipitates were DNA probes with FAM linking to a carboxylated protein-modified upconverted nanomaterial fluorescent probe (DNA/dBSA /NaYF_4_:Yb, Tm).

## 3. Experimental Results and Discussion

### 3.1. The Effect of Yb^3+^ Doping Concentration on the Luminescence of Upconverted Nanomaterials

Figure 1 shows the energy structure and the transitions of Yb^3+^ and Tm^3+^ ions in the upconverting luminescence process. It can be seen that upconversion luminescence is a complex multi-photon energy transfer and conversion process. The energy level transition of ^2^F_7/2_ → ^2^F_5/2_ of the sensitizer Yb^3+^ ion matches the energy of the near-infrared photon at 980 nm, so it can continuously absorb the excitation energy and then transfer it to the adjacent luminescent center Tm^3+^. The ^3^H_5_, ^3^F_2_ (^3^F_3_), and ^1^G_4_ energy levels are from Tm^3+^ ions. Among them, there are three methods for upconversion luminescence: (1) the ^3^H_6_ energy level absorbs three photons continuously and transitions to the ^1^G_4_ energy level; (2) the ^3^H_6_ energy first absorbs two photons continuously and then transitions to the ^3^F_2_, then, through the cross-relaxation process ^3^F_2,3_ + ^3^H_4_ → ^3^H_6_ + ^1^D_2_ to the ^1^D_2_; (3) the non-radiation transitions from ^1^G_4_, ^1^D_2_ of Tm^3+^ to the lower energy levels ^3^F_2,3,4_, ^3^H_4,5,6_ to achieve upconversion luminescence. From the emission spectra of NaYF_4_:Yb,Tm and NaGdF_4_:Yb,Tm in Figure 3b,d, it can be seen that different host materials NaY(Gd)F_4_ will not affect the position of the emission peak, but the doping concentration of Yb^3+^ ions affects the intensity of the emission peak. However, the change is not a simple linear increase or decrease with the increase or decrease in the doping ratio of Yb^3+^.

### 3.2. The Effect of Rare-Earth Ion Doping Concentration on the Luminescence of Upconverted Nanomaterials

#### 3.2.1. Effect of Yb^3+^ Doping Concentration on Upconverted Nanomaterials

NaYF_4_:x%Yb^3+^, 0.5%Tm^3+^ and NaGdF_4_:x%Yb^3+^, 0.5%Tm^3+^ (x = 5, 10, 20, 50, 80) nanomaterials were synthesized by high-temperature thermal decomposition under the same experimental conditions by varying the doping molar fraction of Yb^3+^ ions with a fixed Tm^3+^ ions molar fraction of 0.5%. Figure 2 and Figure 3 show the multi-directional characterization results of the prepared NaYF_4_:x%Yb^3+^, 0.5%Tm^3+^ and NaGdF_4_:x%Yb^3+^, 0.5%Tm^3+^. As shown in Figure 2a–j and Figure 3a–j, all the nanomaterials exhibit the characteristics of high size dispersion and good crystallinity. It can be seen from Figure 2k and Figure 3k that NaYF4: 20%Yb^3+^, 0.5%Tm^3+^ and NaGdF_4_:20%Yb^3+^, 0.5%Tm^3+^ are uniform in morphology and size, forming a complete and regular hexagonal phase. Obviously, the doping concentration of Yb^3+^ ions does not have much effect on the morphology of the upconverted nanomaterials. The nanomaterials with different Yb^3+^ doping ratios can be synthesized with particle sizes between 25–38 nm, which lays a good foundation for the subsequent preparation of upconversion fluorescent probes. NaYF_4_:x%Yb^3+^, 0.5%Tm^3+^(x = 5, 10, 20, 50, 80) and NaGdF_4_:x%Yb^3+^, 0.5%Tm^3+^ (x = 5, 10, 20, 50, 80) were subjected to phase analysis, as shown in Figure 2l and Figure 3l, the XRD patterns were compared with standard card No.16-0994 (NaF_4_), and the diffraction peaks obtained all corresponded to the standard card one by one, indicating that the samples obtained under this condition were all pure hexagonal NaYF_4_:Yb, Tm and NaGbF_4_:Yb, Tm.

**Figure 2 nanomaterials-12-01787-f002:**
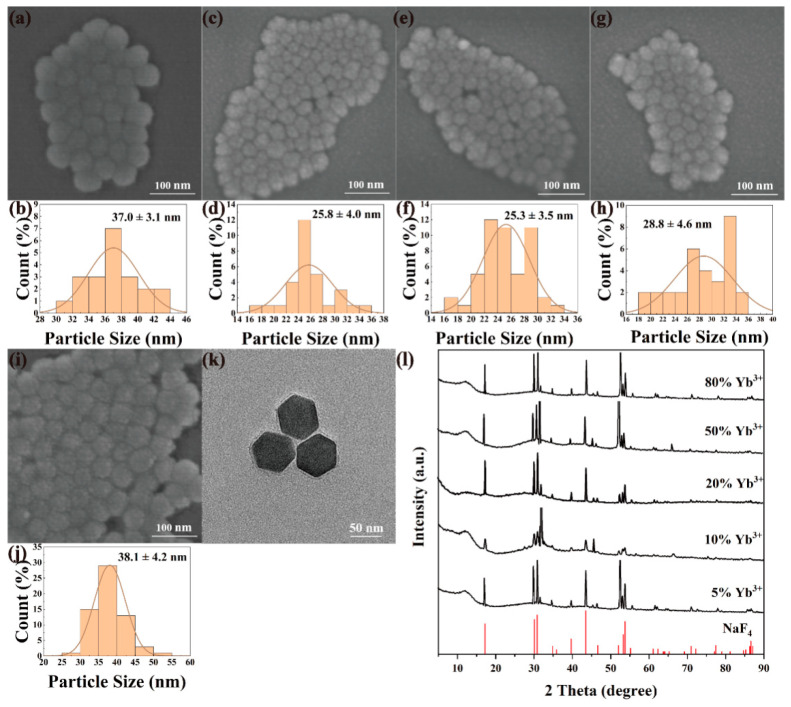
SEM, particle size analysis, TEM and XRD images of NaYF_4_: x%Yb^3+^, 0.5%Tm^3+^. SEM: (**a**) x = 5, (**c**) x = 10, (**e**) x = 20, (**g**) x = 50, (**i**) x = 80; particle size analysis: (**b**) x = 5, (**d**) x = 10, (**f**) x = 20, (**h**) x = 50, (**j**) x = 80; (**k**) TEM image of NaYF_4_: 20%Yb^3+^, 0.5%Tm^3+^; (**l**) XRD of NaYF4:x%Yb^3+^,0.5%Tm^3+^ (x = 5, 10, 20, 50, 80).

**Figure 3 nanomaterials-12-01787-f003:**
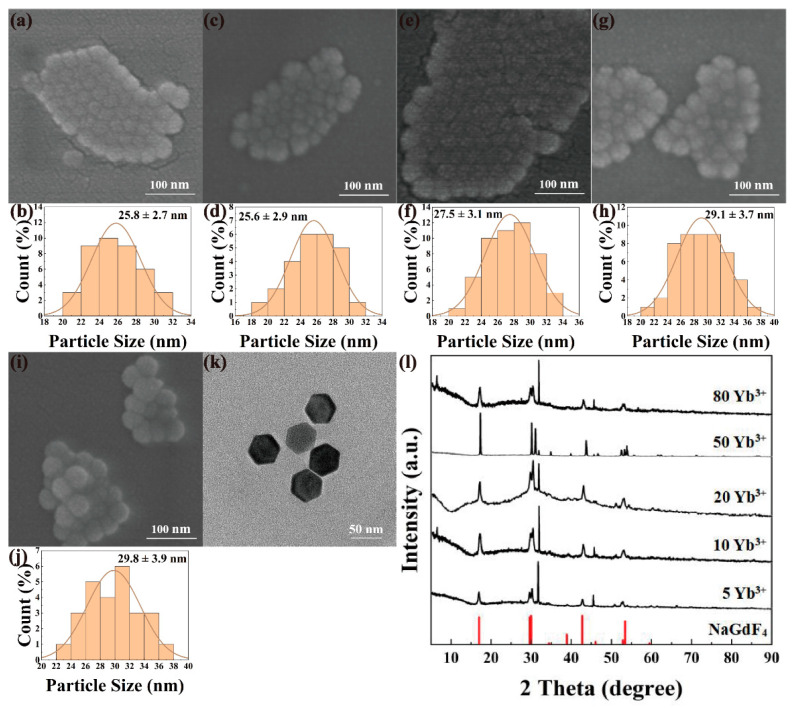
SEM, particle size analysis, TEM and XRD images of NaGdF_4_: x%Yb^3^^+^, 0.5%Tm^3^^+^. SEM: (**a**) x = 5, (**c**) x = 10, (**e**) x = 20, (**g**) x = 50, (**i**) x = 80; particle size analysis: (**b**) x = 5, (**d**) x = 10, (**f**) x = 20, (**h**) x = 50, (**j**) x = 80; (**k**) TEM image of NaGdF_4_: 20%Yb^3+^, 0.5%Tm^3+^; (**l**) XRD of NaGdF_4_:x%Yb^3+^, 0.5%Tm^3+^ (x = 5, 10, 20, 50, 80).

Figure 4 shows the fluorescence spectra of nanomaterials with different concentrations of Yb^3+^ doping. It can be seen from Figure 4a,c that the emission peak position is not affected by the host material NaY(Gd)F_4_ or the doping concentration of Yb^3+^ ions. The intensity of the emission peak changes as the doping ratio of Yb^3+^ changes, but the change is not a simple linear increase or decrease with the increase or decrease in the doping ratio of Yb^3+^.

As shown in Figure 4, the luminescence intensity at 450, 477 and 646 nm increased gradually with the increase in Yb^3+^ concentration up to 20%, then the emission intensity is decreased with the increase in Yb^3+^ concentration (Figure 4b,d).

Upon Yb^3+^ doping, with the change in doping concentration, the number of photons absorbed at 980 nm increases, and the energy transferred to Tm^3+^ ions increases, so that its luminescence is enhanced. When the concentration of Yb^3+^ continues to increase, the photon energy absorbed by the Yb^3+^ ions will pass through the “bridge” between Yb^3+^ − Yb^3+^ and surface defects. Through energy resonance transfer, the energy will be transferred to the surface defects and organic vibration groups, through the free radiation process. According to the experimental results, when the optimal Yb^3+^ doping mole fraction is 20%, the luminescence reaches its peak.

#### 3.2.2. The Effect of Tm^3+^ Doping Concentration on Upconverted Nanomaterials

NaYF_4_:20%Yb^3+^, x%Tm^3+^ and NaGdF_4_:20%Yb^3+^, x%Tm^3+^ were synthesized by the same method and conditions by varying the concentration of Tm^3+^ (x = 0.2, 0.3, 0.5, 0.8, 1.0) while the concentration of Tb^3+^ was fixed at 20%. Figure 5 and Figure 6 show the multi-directional characterization results of the prepared NaYF_4_:20%Yb^3+^, x%Tm^3+^ and NaGdF_4_:20%Yb^3+^, x%Tm^3+^. As can be seen in Figure 5a–j and Figure 6a–j, all the nanomaterials have high dispersibility in size distribution and good crystallinity. It can be seen from Figure 5k and Figure 6k that the upconverted nanomaterials have uniform morphology and size, forming a regular hexagonal phase. Similarly, by changing the doping concentration of Tm^3+^, the morphology of the upconverted nanomaterials does not change too much. The particle sizes of nanomaterials with different Tm^3+^ doping ratios can be between 21–43 nm. The XRD pattern of NaGdF_4_: x%Yb^3+^, 0.5%Tm^3+^ (x = 0.2, 0.3, 0.5, 0.8, 1) and NaGdF_4_: 20%Yb^3+^, x%Tm^3+^ (x = 0.2, 0.3, 0.5, 0.8, 1) nanomaterials was also compared with the standard card No.27-0699 (NaGdF_4_), NaGdF_4_: Yb, Tm is a pure hexagonal phase. As shown in Figure 5l and Figure 6l, the diffraction peaks obtained all corresponded to the standard card, which means that the samples obtained under this condition were all pure hexagonal NaYF_4_:Yb, Tm and NaGbF_4_:Yb, Tm.

The luminescence spectra of materials with different Tm^3+^ concentrations are shown in Figure 7. As the mole fraction of Tm^3+^ increases from 0.2% to 0.5%, the luminescence peaks at 450, 475, and 646 nm also change. As the mole fraction of Tm^3+^ increases from 0.5% to 1%, the emission intensities of these three luminescence peaks gradually decrease. When the mole fraction of Tm^3+^ is 0.2%, the luminescence intensity of the nanomaterial is not strong because there are not enough excitable Tm^3+^ ions in the nanomaterials. As the concentration of Tm^3+^ increases gradually, the number of excitable Tm^3+^ ions increases accordingly, and the luminescence of nanomaterials becomes stronger accordingly. When the mole fraction of Tm^3+^ reaches 0.5%, the upconversion luminescence intensity reaches the maximum, and then gradually becomes weaker. This is because the increase in Tm^3+^ ion concentration reduces the interionic distance and strengthens the interaction. Finally, the concentration quenching effect and cross-relaxation effect are observed, resulting in a decrease in luminescence intensity. At the same time, when the sample is excited with the same power of the 980 nm laser, since the total energy is fixed, the energy that each Tm^3+^ can receive will decrease with the increase in Tm^3+^, leading to the weakening of the luminescence.

#### 3.2.3. Comparison of Luminescence Properties of Two Upconverted Nanomaterials with the Best Doping Ratio

From the above results, the luminescence intensity of nanomaterials is the strongest when the mole fraction of Yb^3+^ is 20 % and the mole fraction of Tm^3+^ is 0.5 %. Therefore, the optimal doping concentration ratio (40:1) is selected to prepare these two kinds of upconverting nanomaterials. Figure 8 shows the luminescence intensity comparison of NaYF_4_:20%Yb^3+^,0.5%Tm^3+^ and NaGdF_4_:20%Yb^3+^,0,5%Tm^3+^. At 345, 362, 450, 477, 646, 802 nm, the luminescence intensity of NaYF_4_:Yb,Tm is 4.4, 3.0, 4.2, 3.4, 2.3, and 2.7-times stronger than that of NaGdF_4_:Yb,Tm, respectively. The luminescence intensity of NaYF_4_:Yb, Tm is much higher than that of NaGdF_4_:Yb, Tm under the same power of 980 nm laser excitation. Therefore, we chose NaYF_4_:20%Yb^3+^,0.5%Tm^3+^ upconverted nanomaterials as the substrate to further investigate protein detections.

### 3.3. Analysis of Fluorescence Characteristics Based on NaYF_4_:Yb^3+^, Tm^3+^ Biological Probes

After the carboxylated bovine serum albumin was added to the upconverted nanomaterial sample for reaction treatment, we carried out repeated centrifugal washing on the sample to remove the carboxylated bovine serum albumin, and then measured the infrared absorption spectrum of the remaining samples, as shown in Figure 9a (the red line). As shown in Figure 9a, the broad infrared (IR) absorption peak at 3295.86 cm^−1^ represents the stretching vibration peak of OH in the carboxyl functional group; the sharp IR absorption peak at 1650.82 cm^−1^ represents the C=O formed after the reaction between the carboxylated protein and NH_2_-PEG group stretching vibration peak; 1024.05 cm^−1^ represents the stretching vibration absorption peak of -O- in PEG; 700.06 cm^−1^ broad absorption peak represents the out-of-plane rocking vibration absorption peak of NH in bovine serum albumin and NH_2_-PEG. Repeated centrifugal washing can completely remove the free carboxylated bovine serum albumin in the solution, and the upconversion material is inorganic and will not absorb at these positions. Therefore, it is considered that the new characteristic peaks belonging to organic functional groups can only come from carboxylated bovine serum proteins that have been attached to the surface of upconverted nanomaterials, which cannot be washed away. The above results prove that the surface of NaYF_4_:Yb^3+^, Tm^3+^ nanomaterials contains many carboxyl functional groups, and the carboxylated bovine serum albumin has been successfully modified the surface of water-soluble nanomaterials. Figure 9b shows the excitation and emission spectra of DNA/dBSA/NaYF_4_:Yb,Tm excited at 480 nm. The excitation spectrum is from 450 nm to 490 nm, and the emission spectrum is from 510 nm to 530 nm, which is mainly the contribution of FAM. As described in Section 2.6, we carried out repeated centrifugal washing to fully wash the excess DNA, and then obtained the fluorescence spectrum in Figure 9c. As shown in Figure 9c, when the DNA/dBSA/NaYF_4_:Yb,Tm fluorescent probes were excited at 480 nm, a strong emission peak is observed at 520 nm, which is consistent with the FAM fluorescence peak. It indicates that some DNA strands were not washed away due to their attachment to the carboxylated protein-modified upconversion material, namely DNA was attached successfully to the surfaces of UCNP via a protein and a FAM was added to the UCNP/DNA complex. Similarly, NaYF_4_:Yb,Tm and DNA/dBSA/NaYF_4_:Yb,Tm was excited at 980 nm, and the upconversion intensity of the luminescence is quite strong, as shown in Figure 9d. These nanocomposites have strong upconversion luminescence with good water solubility and biocompatibility, making them a new type of fluorescent probe.

### 3.4. Fluorescent Probes for the Detection of Different Proteins

To test the detection of these upconverted nanomaterials for protein detection, 100 pM solutions of miRNA-155, single-base mismatch of miRNA-155, double-base mismatch of miRNA-155, complete-base mismatch of miRNA-155 and miRNA-150 solution with the same concentration of 100 pM were prepared, respectively. The prepared nucleotide sequences of miRNAs and fluorescent probes are shown in Table 1. The prepared fluorescent probes of upconverted nanomaterials were tested with different miRNAs and mismatched miRNAs, and the samples were excited by a fiber laser with a wavelength of 980 nm and their fluorescence spectra were measured.

As shown in Figure 10, the fluorescence spectrum of the fluorescent probe after connecting different miRNA-155 changed significantly. In general, the fluorescence spectra of the four groups were very similar, and the fluorescence intensity decreased significantly (compared with Figure 9d). The sample added with miRNA-155 had the strongest fluorescence intensity. The more mismatched bases, the more obvious fluorescence quenching and the smaller spectral intensity. It is worth noting that the fluorescence quenching at 345 nm, 362 nm, 450 nm, 477 nm and 646 nm is more obvious than that at 802 nm.

We divided the peak intensity at 802 nm (*I*_802_) by the peak intensity at 345 nm (*I*_345_), 362 nm (*I*_362_), 450 nm (*I*_450_), 477 nm (*I*_477_) and 646 nm (*I*_646_) to calculate a group of fluorescence peak ratios for further analysis of the differences in fluorescence spectra of samples with different miRNA-155s. The results are shown in Table 2. As can be seen in Table 2, the five peak ratios of the fluorescent probes are very close to those of the upconverted nanomaterials; then, the value of the completely mismatched miRNA155 is relatively close to that of the upconverted nanomaterials, and intact miRNA-155 had the greatest effect on all five peak ratios. It seems that the completely mismatched miRNA-155 has little effect on the peak ratios, and the intact miRNA-155 has the greatest effect on the peak ratios. This result may be that miRNAs with different sequences have different effects on different molecular bonds of upconverted nanomaterials. It is believed that these peak ratios can be used for specific recognition of miRNA-155. For fluorescent probes with multiple emission peaks, in addition to identifying the target substance by simply comparing the changes in peak intensity, the ratios between peaks can also be used for substance-specific identification. The fluorescence probe with multiple emission peaks provides more abundant optical information for the study of the characteristic changes of the detected object and has great application potential.

As shown in Figure 11, when the fluorescent probes were connected to miRNA-155 and miRNA-150, respectively, the fluorescence intensity of miRNA-155 was higher than that of miRNA-150. The experimental results show that the fluorescent probe can effectively distinguish different types of miRNAs.

## 4. Conclusions

In this paper, a novel NaYF_4_:Yb,Tm surface-functionalized fluorescent probe was proposed based on upconverted nanomaterials. Quantitative analysis of the effects of Yb^3+^ and Tm^3+^ ion concentrations on the morphology, size, and luminescence properties of NaYF_4_:Yb,Tm and NaGdF_4_:Yb,Tm indicated that the optimal doping concentration ratio of Yb^3+^:Tm^3+^ is 40: 1. In the study, by comparing the fluorescence emission of NaYF_4_: 20% Yb, 0.5% Tm and NaGdF_4_: 20% Yb, 0.5% Tm, upconverted nanomaterials with better luminescence properties were obtained. A novel fluorescent probe was designed for the surface carboxylation of NaYF_4_:Yb,Tm and the connection with amino group and DNA. The probe can be used for the detection of different specific biological miRNAs. When the fluorescent probes are used to detect different miRNAs, they can distinguish different miRNAs, especially miRNAs with base mismatches, from non-specific RNA molecular analytes. The preliminary studies indicate that the upconverted probes have a good potential for protein detection in early cancer diagnosis.

## Figures and Tables

**Figure 1 nanomaterials-12-01787-f001:**
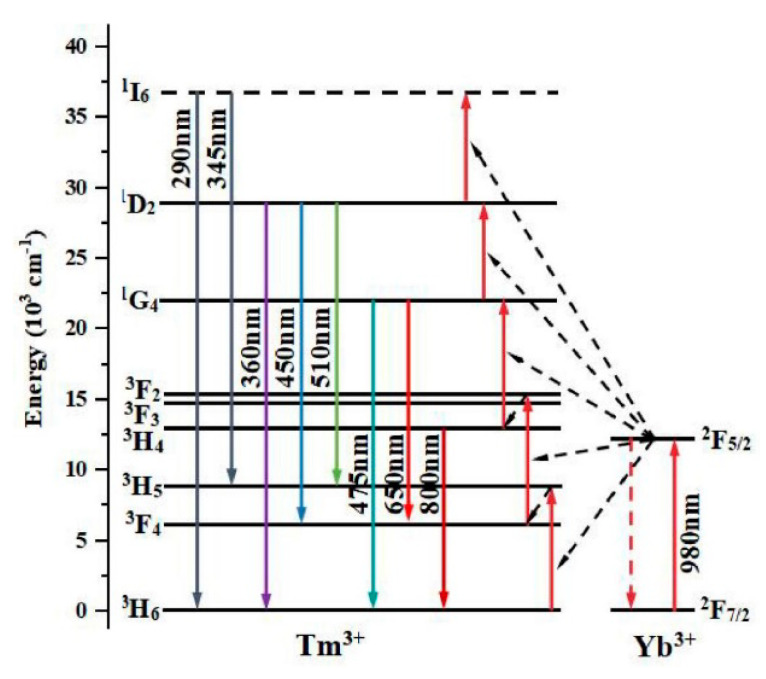
Schematic diagram of energy level transition of Yb^3+^ and Tm^3+^ ions.

**Figure 4 nanomaterials-12-01787-f004:**
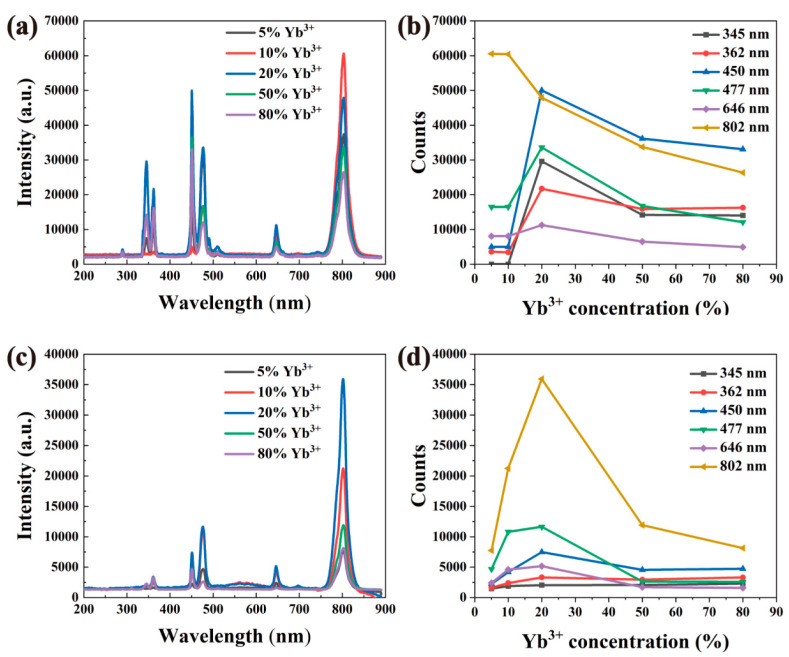
Fluorescence spectra of NaYF_4_: Yb^3+^, Tm^3+^ and NaGdF_4_: Yb^3+^, Tm^3+^ with different Yb^3+^ doping ratios. (**a**) Fluorescence spectra of NaYF_4_:x%Yb^3+^,0.5%Tm^3+^ (x = 5, 10, 20, 50, 80) at 980 nm excitation. (**b**) Schematic diagram of the relationship between the maximum intensity of the luminescence peaks corresponding to NaYF_4_: Yb^3+^ and Tm^3+^ with different doping ratios and the concentration change. (**c**) Fluorescence spectra of NaGdF_4_:x%Yb^3+^,0.5%Tm^3+^ (x = 5, 10, 20, 50, 80) at 980 nm excitation. (**d**) Schematic diagram of the relationship between the maximum intensity of the luminescence peaks corresponding to NaGdF_4_: Yb^3+^ and Tm^3+^ with different doping ratios and the concentration changes.

**Figure 5 nanomaterials-12-01787-f005:**
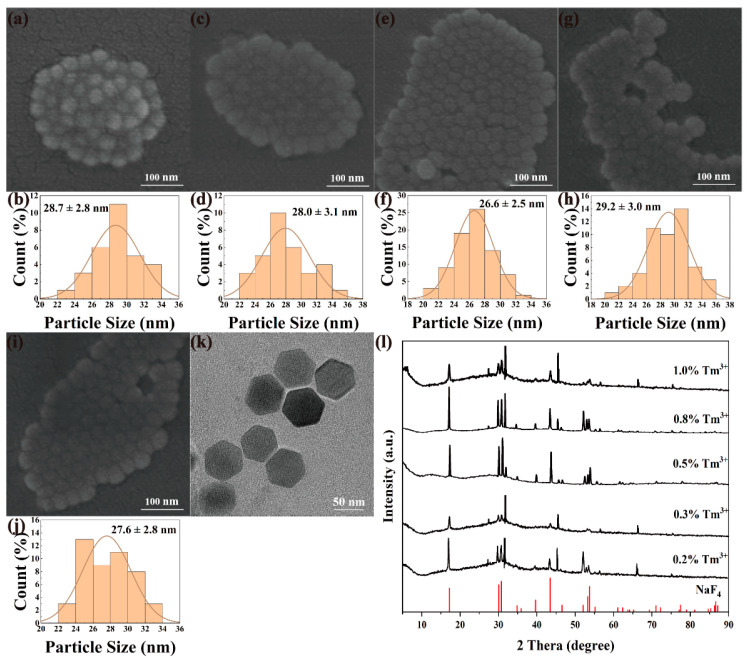
SEM, particle size analysis, TEM, and XRD images of NaYF_4_: 20%Yb^3+^, x%Tm^3+^. (**a**) x = 0.2, (**c**) x = 0.3, (**e**) x = 0.5, (**g**) x = 0.8, (**i**) x = 1.0; particle size analysis: (**b**) x = 0.2, (**d**) x = 0.3, (**f**) x = 0.5, (**h**) x = 0.8, (**j**) x = 1.0; (**k**) TEM image of NaYF_4_: 20%Yb^3+^, 0.5%Tm^3+^; (**l**) NaYF_4_: 20%Yb^3+^, x%Tm^3+^ (X = 0.2, 0.3, 0.5, 0.8, 1.0).

**Figure 6 nanomaterials-12-01787-f006:**
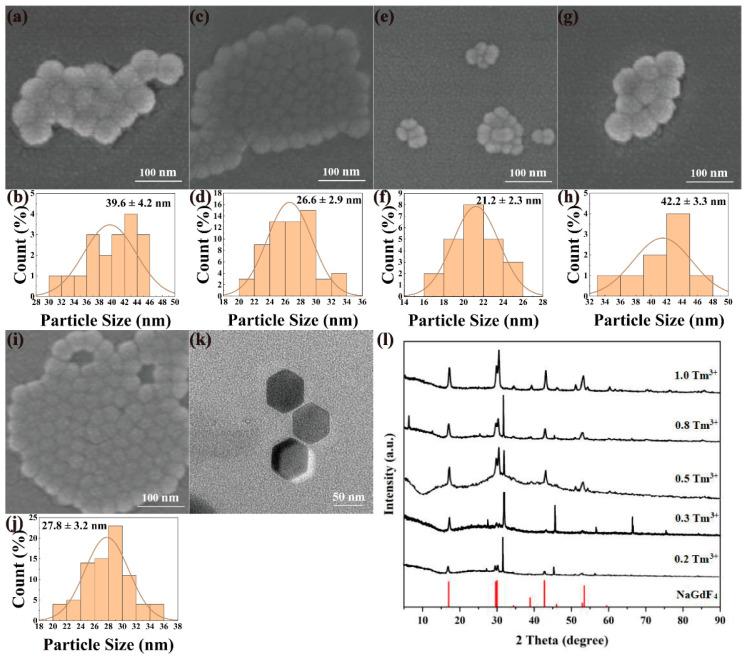
SEM, particle size analysis, TEM, and XRD images of NaGdF_4_: 20%Yb^3+^, x%Tm^3+^. (**a**) x = 0.2, (**c**) x = 0.3, (**e**) x = 0.5, (**g**) x = 0.8, (**i**) x = 1.0; particle size analysis: (**b**) x = 0.2, (**d**) x = 0.3, (**f**) x = 0.5, (**h**) x = 0.8, (**j**) x = 1.0; (**k**) TEM image of NaGdF_4_: 20%Yb^3+^, 0.5%Tm^3+^; (**l**) NaGdF_4_: 20%Yb^3+^, x%Tm^3+^ (X = 0.2, 0.3, 0.5, 0.8, 1.0).

**Figure 7 nanomaterials-12-01787-f007:**
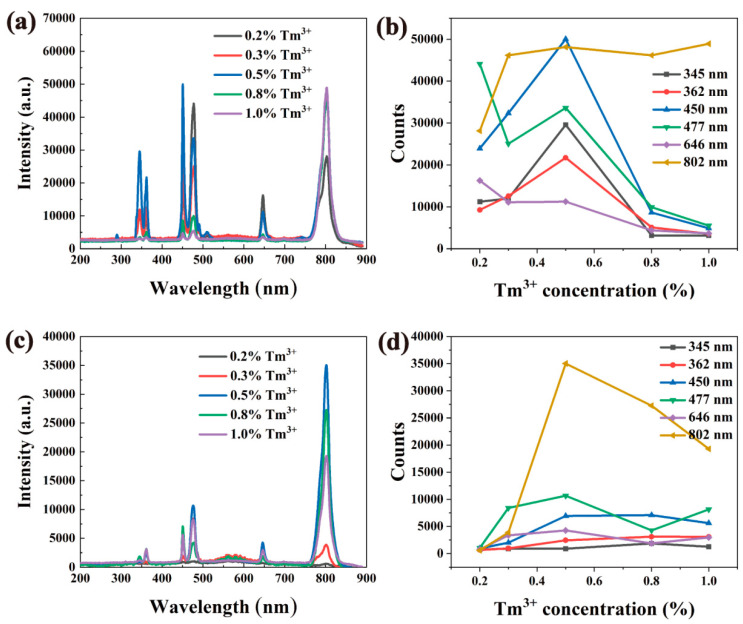
Fluorescence spectra of NaYF_4_: Yb^3+^, Tm^3+^ and NaGdF_4_: Yb^3+^, Tm^3+^ with different Tm^3+^ doping ratios. (**a**) Fluorescence spectra of NaYF_4_:20%Yb^3+^, x%Tm^3+^ (x = 0.2, 0.3, 0.5, 0.8, 1.0) at 980 nm excitation. (**b**) NaYF_4_: Yb^3+^ with different Tm^3+^ doping ratios, the relationship between the maximum intensity of the luminescence peak corresponding to Tm^3+^ and the concentration change. (**c**) Fluorescence spectra of NaGdF_4_:20%Yb^3+^, x%Tm^3+^ (x = 0.2, 0.3, 0.5, 0.8, 1.0 at 980 nm excitation). (**d**) The corresponding doping ratios of NaGdF_4_: Yb^3+^ and Tm^3+^ with different Tm^3+^ doping ratios.

**Figure 8 nanomaterials-12-01787-f008:**
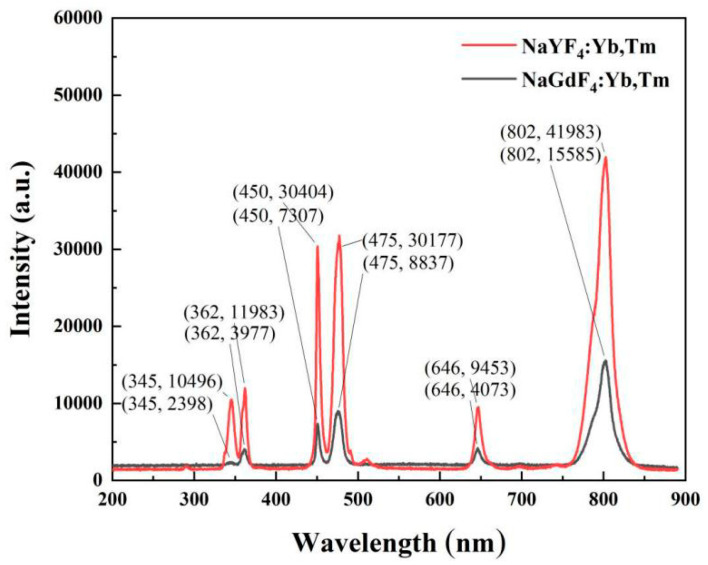
Comparison of fluorescence spectra of NaYF_4_: 20% Yb^3+^, 0.5% Tm^3+^ and NaGdF_4_: 20% Yb^3+^, 0.5% Tm^3+^ at 980 nm excitation.

**Figure 9 nanomaterials-12-01787-f009:**
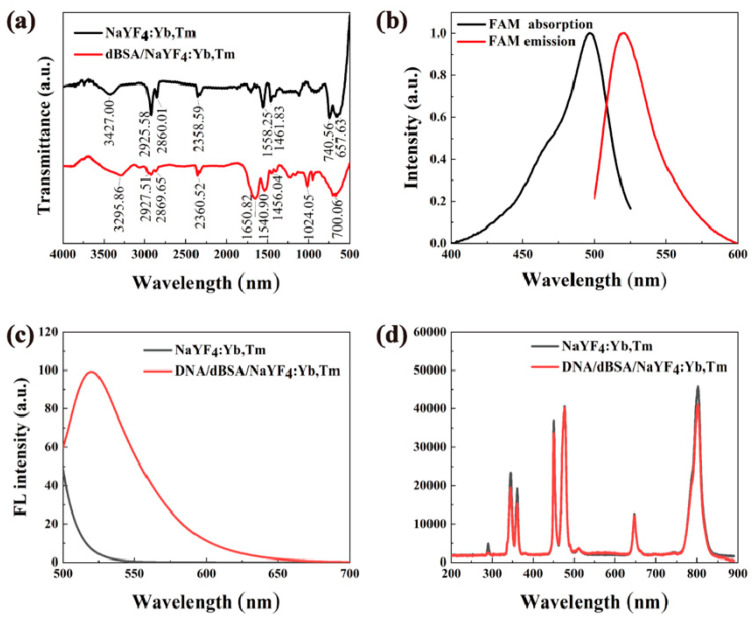
(**a**) Infrared absorption spectra of NaYF_4_: Yb,Tm before and after modification with carboxylated bovine serum albumin. (**b**) Absorption and emission spectra of the FAM fluorophore. (**c**) Fluorescence spectra of NaYF_4_:Yb,Tm and novel fluorescent probes under excitation at 480 nm. (**d**) Fluorescence spectra of NaYF_4_:Yb,Tm and novel fluorescent probes under excitation at 980 nm.

**Figure 10 nanomaterials-12-01787-f010:**
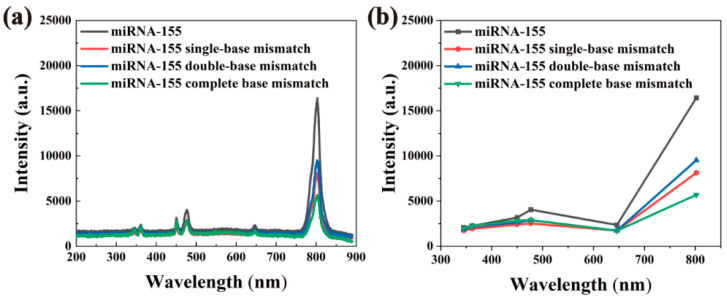
Fluorescence spectra of fluorescent probes connected to miRNA-155, miRNA-155 single-base mismatch, miRNA-155 double-base mismatch, and miRNA-155 complete-base mismatch. (**a**) Fluorescence spectra of mismatched sequences relative to miRNA-155 at 980 nm excitation. (**b**) MiRNA-155 with different sequences, the relationship between the maximum intensity of the luminescence peak corresponding to sequences change.

**Figure 11 nanomaterials-12-01787-f011:**
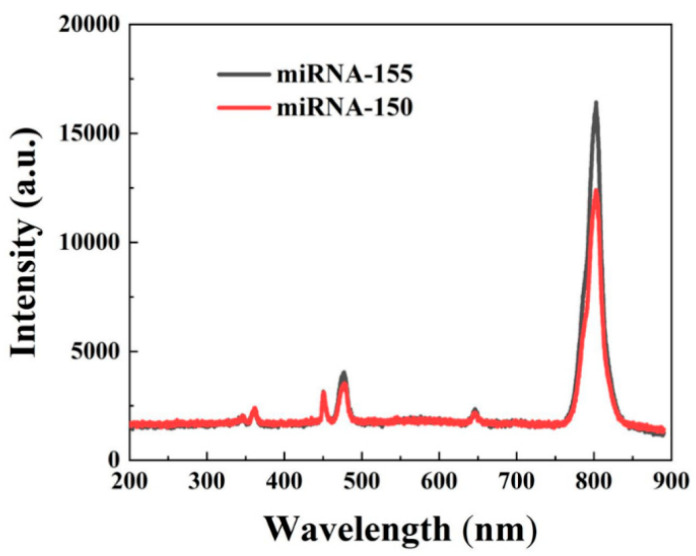
Fluorescence spectra of fluorescent probes connected to miRNA-155 and miRNA-150 under excitation at 980 nm.

**Table 1 nanomaterials-12-01787-t001:** Nucleotide sequences of miRNAs and fluorescent probes.

Name	Sequences (5′-3′)
miRNA-155	UUAAUGCUAAUCGUGAUAGGGGU
miRNA-150	UCUCCCAACCCUUGUACCAGUG
miRNA-155 matched DNA strands	NH_2_-CCCCCCCCCCCC-ACCCCTATCACGATTAGCATTAA-CGCTAT-FAM
miRNA-150 matched DNA strands	NH_2_-CCCCCCCCCCCC-CACTGGTACAAGGGTTGGGAGA-CGCTAT-FAM
miRNA-155 single base mismatch	UUAAGGCUAAUCGUGAUAGGGGU
miRNA-155 double base mismatch	UUAAGGCUAAUAGUGAUAGGGGU
miRNA-155 complete base mismatch	AATTACGATTAGCACTATCCCCA

**Table 2 nanomaterials-12-01787-t002:** The ratio of fluorescence peaks at 802 nm and 450 nm.

Fluorescent Substance	*I*_802_/*I*_345_	*I*_802_/*I*_362_	*I*_802_/*I*_450_	*I*_802_/*I*_477_	*I*_802_/*I*_646_
NaYF_4_:20%Yb^3+^, 0.5%Tm^3+^	1.97	2.38	1.24	1.13	3.65
Fluorescent probes	2.09	2.71	1.23	1.01	3.37
FP + CmiRNA-155	2.87	2.52	2.01	1.96	3.32
FP+ M2miRNA-155	5.19	4.38	3.61	3.29	5.5
FP + M1miRNA-155	4.66	4.21	3.38	3.22	4.69
FP + miRNA-155	7.92	7.28	5.19	4.06	6.99

Noting: FP + CmiRNA-155 represents fluorescent probe with completely mismatched miRNA155; FP+ M2miRNA-155 represents fluorescent probe with miRNA-155 double-base mismatch: FP+ M2miRNA-155 represents fluorescent probe with miRNA-155 single-base mismatch; FP + miRNA-155 represents fluorescent probe with miRNA-155.

## Data Availability

Not applicable.

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
