# Peer review of "Novel Fluorescent Probe Based on Rare-Earth Doped Upconversion Nanomaterials and Its Applications in Early Cancer Detection"

_nanomaterials, 2022, doi:10.3390/nano12111787_

Round 1

Reviewer 1 Report

The paper is devoted to the preparation of the UC-based probe for protein detection. Despite the results are interesting, and can be potentially published, the paper drastically lacks analysis. What is presented is mainly luminescence data and a scheme + some TEM images, while the compound, which was prepared, is a complex object, and prior to publication its characterization on every step of the preparation must be carefully run and presented: starting from the preparation of the initial fluoride (PXRD and EDX are the least to be performed), then to carboxylated protein, then to its attachment (the proof of attachment? the layer thickness? ....), and then DNA and DNA linking. Before all of this is presented, the paper cannot be published. 

Some minor comments also include:

  • title: "their applications ????? protein detection" - something is missing
  • fluorescence is a process of emission with conserved multiplicity, which is not the Tm case. Substitute with luminescence everywhere.
  • "based on a new type of up-conversion nanomaterials NaYF4:Yb,Tm" - this is an old type, please rephrase to make clear what did you mean
  • "the Yb3+/Er3+ (Tm3+ , Ho3+) pairs are recognized as the most efficient" - what was the optimal metal ratio? Why was it important to re-determine it? Is there a correlation between literature and new data?

Reviewer 2 Report

The entire manuscript needs intensive language editing

Reviewer 3 Report

The authors have reported on the use of NaYF4:Yb,Tm fluorescent probe in cancer diagnosis applications. This manuscript could be of interest to the readers of this journal. However, there are a few issues that need to be addressed before suggesting for publication.

  • Overall English of the paper needs to be improved.
  • “Novel fluorescent probes prepared from rare earth-doped up-conversion nanomaterials and their applications protein detection for early cancer diagnosis”, the title is not grammatically correct.
  • The cancer diagnosis mechanism is rather ambiguous and needs to be supported by relevant experiment/result.
  • The introduction needs to be further strengthened by adding relevant references, for example in the sentence below: “In the current practical application fields of biomarkers and bio-detection, up conversion nanoparticles show unique and great advantages compared with traditional organic dyes, quantum dots (doi.org/10.1002/advs.201700548), fluorescent proteins, and other biomolecular markers [32-34], such as near-infrared light as excitation light, resulting in less photo-damage, lower auto-78 fluorescence background of biological tissue, and a deep penetration depth.” Mentioned refences could be included.
  • Average particle size distribution of the synthesized nanoparticles needs to be measured and included to the Figure 2.

Round 2

Reviewer 1 Report

My comments were taken care of. The paper can be accepted.

Reviewer 3 Report

The authors have addressed all the comments mentioned before. The manuscript is suggested for publication in this journal.